# Double Fatal Sodium Nitrite Poisoning—Double Homicide, Extended Suicide, or Suicide and Accident?

**DOI:** 10.3390/ijms27010218

**Published:** 2025-12-24

**Authors:** Anna Smędra, Katarzyna Wochna, Mateusz Lisowski, Agnieszka Skulska-Birgiel, Marta Suchan, Jarosław Berent

**Affiliations:** 1Department of Forensic Medicine, Medical University of Lodz, Urzednicza 44, 91-304 Lodz, Poland; anna.smedra@umed.lodz.pl (A.S.); jaroslaw.berent@umed.lodz.pl (J.B.); 2Toxicology Department, Clinic of Anesthesiology and Intensive Care, Central Clinical Hospital, Medical University of Lodz, Pomorska 251, 92-213 Lodz, Poland; mlisowski@csk.umed.pl; 3Sehn Institute of Forensic Research, Westerplatte 9, 31-033 Krakow, Poland; askulska@ies.gov.pl (A.S.-B.); msuchan@ies.gov.pl (M.S.)

**Keywords:** accident, extended suicide, homicide, manner of death, methemoglobin, poisoning, sodium nitrite, suicide

## Abstract

Recent years have seen an increasing number of cases of intentional poisoning with sodium nitrite. This is due, in part, to its inexpensive and readily available nature, and to the growth of social media. This paper briefly reviews the toxicity of sodium nitrite and the mechanism of death associated with its consumption and then presents a case of the fatal sodium nitrite poisoning of two sisters within a few hours of each other. It goes on to discuss both the growing problem of mental disorders, particularly among young people, and the potential diagnostic difficulties associated with sodium nitrite poisoning. It also addresses the approach to determining the manner of death, which may not be evident to the investigator; an accurate investigation is, of course, essential, as mistakes can result in the failure to identify the actual perpetrators and even lead to false accusations of guilt.

## 1. Introduction

Sodium nitrite (NaNO_2_) is a white-yellowish powder with a salty taste, which is readily soluble in water. It has several industrial applications, including its widespread use as a preservative (E250) in food to prevent the growth of anaerobic bacteria (including *C. botulinum*). It is readily available over-the-counter and can be easily purchased both as a curing salt, i.e., mixed with table salt, and as a pure chemical reagent. Studies show that regular consumption of even properly cured meat is not associated with acute toxicity, albeit an increased risk of colon carcinogenesis; however, pure sodium nitrite presents a significant risk to health and life, with the lethal dose for an adult being approximately 1 to 2.5 g [1].

While cases of unintentional, i.e., accidental, nitrite poisoning have been described for at least 80 years [2,3,4,5], clinical practice indicates an increasing number of intentional suicidal poisonings in recent years. Similarly, while few publications have addressed nitrite poisoning, their number increased during the COVID-19 pandemic, and have examined cases in various regions including Asia, Australia, Europe, and North America [6,7,8,9], underlining that this is a global phenomenon. Therefore, it is crucial to increase awareness of nitrite poisoning among physicians. Clinicians should be prepared to make a correct diagnosis and promptly implement appropriate treatment, both at the stage of prehospital care and in hospital emergency departments; in addition, forensic pathologists should be ready to consider the possibility of poisoning, particularly when no identifiable traumatic or disease-related cause of death is apparent during autopsy. To the unprepared, nitrite poisoning may not be immediately easy to confirm: while the body may present a chocolate color of postmortem lividity and blood [10], this is not always true, as demonstrated in the present case.

First, it is important to explain why nitrites pose a threat to human health and life. Briefly, sodium nitrite consumption results in the intense induction of methemoglobinemia (MetHb). Under physiological conditions, oxygen in the form of a superoxide ion (O_2_^−^) binds to the iron atom at the center of the heme unit, temporarily oxidizing the iron to Fe^3+^; when the blood reaches the target tissue, oxygen is released and the heme iron is reduced to Fe^2+^. However, following exposure to reactive oxygen species (ROS) and certain xenobiotics, such as nitrites and nitrates, the iron atom can be permanently oxidized to the Fe^3+^ state. In this situation, the molecule loses its ability to bind oxygen, and instead forms methemoglobin, which is unable to distribute oxygen to tissues.

Under conditions of general oxidative stress, small amounts of methemoglobin are typically present in the blood, and mechanisms have developed to regenerate it into hemoglobin. Erythrocytes can produce NADH-cytochrome b_5_ reductase, which catalyzes the transfer of electrons from NADH, derived from glycolysis, to cytochrome b_5_; this, in turn, reduces methemoglobin to hemoglobin. Under typical conditions, this is essentially the only pathway that can reduce iron atoms in heme molecules. The enzyme is capable of maintaining methemoglobin concentrations in the blood at 1–2% [11]. This process is illustrated in Figure 1.
Figure 1Spontaneous formation of methemoglobin molecules and their reduction to hemoglobin under physiological conditions (green arrow); Blue arrow represents a pathway inactive during normal conditions.However, in conditions of sodium nitrite poisoning, the process of methemoglobin formation exceeds the reducing capacity of the cytochrome b_5_ pathway. Therefore, a second metabolic pathway, NADPH-dependent methemoglobin reductase, is activated. In this pathway, NADPH derived from the pentose phosphate pathway transfers an electron to Fe^3+^, thus restoring functional hemoglobin; in the process, the cofactor methylene blue forms leucomethylene blue [11]. This process is shown in Figure 2.
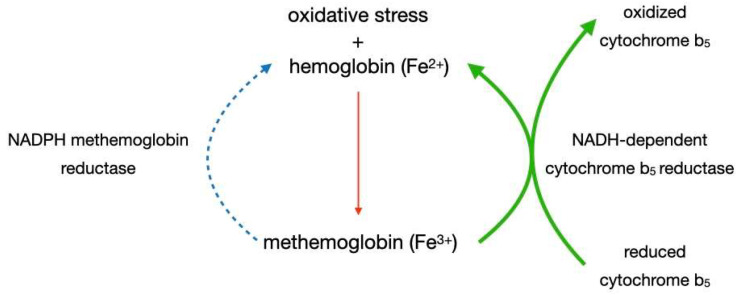

Figure 2Conditions of sodium nitrite poisoning—activation of the second metabolic pathway dependent on NADPH-methemoglobin reductase (blue arrow). The physiological pathway (green arrow) is overwhelmed, thus ineffective.
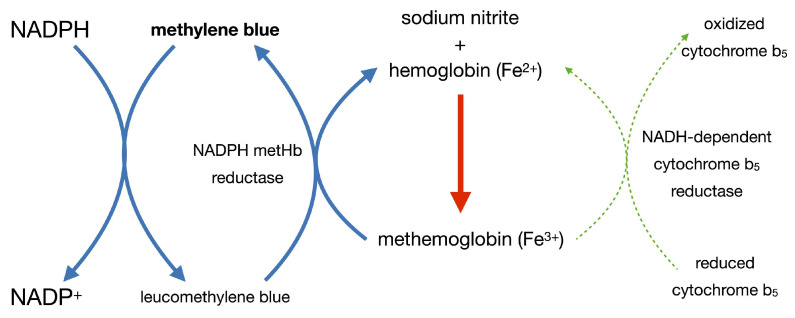


Sodium nitrite poisoning is a life-threatening condition that requires immediate medical attention and appropriate treatment. It is characterized by high concentrations of nitrite ions in blood serum, inducing a biphasic, autocatalytic reaction of hemoglobin oxidation to methemoglobin (MetHb), with the concomitant release of nitric oxide precursors. The reaction of nitrite with hemoglobin results in the production of nitric oxide (II), which exhibits a strong peripheral vasodilatory effect, exacerbating the circulatory and respiratory failure of the poisoned patient.

Patients with MetHb concentrations < 15% are usually asymptomatic, with the exception of poisoned individuals with a history of cardiac or pulmonary disease. At MetHb concentrations > 15%, most patients develop central cyanosis. In some patients, the first objective symptom is a dark brown discoloration of blood drawn for laboratory testing. As MetHb concentrations increase to approximately 30–40%, the patient will demonstrate general weakness, mild disturbances of consciousness, headaches, nausea, tachycardia and tachypnea. Muscle cramps/pain may also occur. Sodium nitrite also has a similar effect on myoglobin; as a result, exposure reduces oxygen transport in the muscles and forces anaerobic metabolism, resulting in the accumulation of lactic acid. MetHb concentrations >50% lead to severe metabolic acidosis and are also associated with symptoms of profound tissue hypoxia, including seizures, coma, and cardiac arrhythmia. Myocardial hypoxia can lead to type II myocardial infarction. Further increases in myoglobin concentration usually lead to death [11].

This paper thoroughly describes the case of sodium nitrite poisoning of a 17-year-old woman and a 31-year-old woman. It is based not only on the autopsy results and toxicology tests, but also on more than 1600 pages of files collected by the prosecutor’s office. In this case, determining the cause of death based on toxicological tests was only a prelude to further steps aimed at assessing the manner of death, which required significant work and the participation of numerous experts.

## 2. Case Report

Around 3:00 a.m., a mother was sleeping in the same room with her two daughters. The mother heard her younger daughter, a 17-year-old, vomiting. Upon checking, she noticed her daughter lying unconscious on the bed in the clothes she had worn the day before: she had not undressed for bed or changed into her pajamas. A bowl containing vomit, mostly liquid with a small amount of food particles, was next to the bed. Her daughter was pale, had blue lips and had bitten her tongue; according to her mother, she was still breathing. The mother called EMS, and an ambulance arrived 10 min later.

In interview, the mother stated that her daughter had not been undergoing any treatment and denied consuming alcohol or psychoactive substances. Signs of cardiopulmonary arrest were observed, with the likely time being 3:05 a.m. The patient’s skin was cool and pale and showed signs of peripheral cyanosis. An ECG revealed signs of PEA. Resuscitation measures were initiated, including indirect cardiac massage and the administration of adrenaline, noradrenaline, atropine and intravenous fluids. After approximately 30 min of resuscitation, a transient return of sinus rhythm was achieved, but PEA recurred approximately 5 min later. During continued resuscitation efforts, the patient was transported to the hospital with her mother. In the emergency room, due to the duration of the cardiac arrest, unsuccessful resuscitation efforts, and certain signs of death, further resuscitation efforts were discontinued, and the patient was pronounced dead at 4:57 a.m. Due to the unknown cause of death, the police and prosecutor’s office were notified, and a decision was made to perform a forensic autopsy; an investigation was initiated for acts under Article 155 of the Polish Penal Code, i.e., involuntary manslaughter.

While the mother was in the hospital with her younger daughter, the older daughter was being cared for by a visually impaired neighbor. The daughter was a 31-year-old woman with autism spectrum disorder and intellectual disability: she did not speak, did not express her basic needs, required constant care from others, and often ate inedible items, such as her own clothes. She was said to be overly agitated and running aimlessly around the apartment. After returning home, the mother took over care of the older daughter. After a few hours, a friend of the mother arrived at the apartment to provide emotional support. The older daughter did not eat all day and went to bed around 1–2 p.m. Around 6:00 p.m., the body of the daughter was discovered, with pale skin and blue lips. A doctor was called, who pronounced the woman dead and reported the incident to the police. A police forensic team arrived at the scene.

A forensic specialist did not rule out the involvement of other people in the death and estimated that it had likely occurred between 9:00 a.m. and 3:00 p.m. A decision was made to perform a forensic autopsy (an investigation into acts under Article 155 of the Polish Penal Code was initiated). Vomit was seized, along with food the daughters allegedly ate the day before: vegan sausages and meatballs, which were past the expiry date. The younger daughter’s phone, laptop, and backpack were also seized, as well as her mother’s phone.

Based on the results of the autopsy and additional tests (alcohol and alcohol-like substance concentrations, histopathology), the forensic pathologists concluded that the cause of death for both women was acute circulatory and respiratory failure of unknown cause. Complete toxicological testing of the material collected during the autopsy was recommended. Additionally, both women were undernourished, with a BMI of 14.5, and the younger sister had scars from self-harm. The color of livor mortis and blood was not typical of nitrite poisoning.

Vomit, sausages/meatballs and autopsy material were sent for further toxicological tests. Tests for alcohol, alcohol-like substances, carbon monoxide, new psychoactive substances, heavy metals, and toxic compounds found in mushrooms were negative. However, positive results were obtained for nitrites in the autopsy material and vomit (Table 1, Table 2 and Table 3). Nitrites were first detected in the vomit using test strips. Subsequently, all recovered material was analyzed for nitrites spectrophotometrically using a Thermo Genesys 10S UV-VIS spectrophotometer (wavelength 525 nm) (Thermo Fisher Scientific, Waltham, MA, USA). In turn, the analysis for nitrates in the obtained filtrates was performed using the semi-quantitative method using Nitrat-Test test strips from Merck (Darmstadt, Germany) (range 10–500 mg/L NO_3_^−^).

In order to isolate the nitrites and nitrates, biological material was deproteinated—blood via zinc sulfate and sodium hydroxide, tissues via sodium borate, potassium ferrocyanide and zinc acetate. Resulting supernatants were analyzed with test strips for presence of nitrates and spectrophotometrically, using Cleve’s acid (blood, derived from Shechter et al. [12]) or sulfanilic acid and alpha-naphtylamine hydrochloride (tissues, derived from Sen et al. [13]).

Blood sample preparation: 1 mL of blood was mixed with 6 mL of zinc sulfate solution and 1 mL of 4% sodium hydroxide solution in a centrifuge tube, then left in ice for 1 h. After centrifugation, 5 mL of supernatant was transferred to a 10 mL volumetric flask, then 0.8 mL of sulfanilic acid was added. The mixture was left in ice for 15 min, after which 0.8 mL of Cleve’s acid (1-aminonaphthalene-6-sulfonic acid) was added. Using deionized water, the solution was diluted to 10 mL and left for 1 h in darkness and room temperature. Absorbance measurements were conducted with the test solution prepared as described.

Tissue sample preparation: 15 g samples of stomach, intestine, kidney and liver were homogenized with 15 mL of distilled water; 5 mL of saturated sodium tetraborate and 50 mL of 70 °C water were added. Mixture was then left in a 100 °C water bath for 15 min. After cooling down to room temperature, 2 mL of 10.6% potassium ferrocyanide solution and 2 mL of 22% zinc acetate solution were added. After 30 min, the deproteinated tissue was transferred to a flask, diluted to 200 mL, and filtered through a filter paper. A 5 mL filtrate sample was transferred to a 25 mL volumetric flask and 10 mL of Griess I + II reagents were added (sulfanilic acid and alpha-napthtylamine). The flask was then filled with water, and after 1 h, the absorbance measurements were conducted.

Obtained solutions were analyzed for nitrites spectrophotometrically using the Genesys 10S UV-VIS manufactured by Thermo. Measurements were conducted using a 525 nm wavelength. The analysis for nitrates was conducted via Nitrat-Test strips manufactured by Merck (range 10–500 mg/L NO^3−^).

Nitrite levels were detected in food products, including sausages and vegetable meatballs, at levels comparable to those found in the reference materials.

After obtaining the results of toxicological tests, it was eventually concluded that both women died from nitrite poisoning. The evidence collected at the time did not indicate the source of the nitrites, except that it could be determined that they were not from the food preserved in the apartment.

After analyzing the material collected at the beginning of the investigation, viz. the inspection of the place of death of the older daughter, witness interviews, and autopsy results of both women, the investigators proposed four possible scenarios: 1. The mother poisoned both daughters; 2. The younger daughter died by suicide, and then the mother poisoned the older daughter; 3. The younger daughter decided to commit suicide and deliberately (intentionally) left sodium nitrite for the older daughter, knowing that the latter often ate “inedible” things (extended suicide); 4. The younger daughter died by suicide, and the older daughter accidentally consumed poison (accident).

During the subsequent investigation, it was determined that the family was experiencing serious financial problems. The mother was unemployed because she had to care for her older daughter. The only source of income was financial support for the children: child-raising allowance and child support for the younger daughter, and an allowance for the older daughter. Sometimes there was nothing to eat at home, and the mother would give her daughters sleeping pills to keep them from feeling hungry. A search of the seized electronic equipment revealed numerous messages indicating that the mother had purchased and consumed drugs and provided them to her younger daughter together with alcohol. For this reason, during the proceedings, she was charged with providing alcohol and drugs to a minor. Furthermore, people around her, including teachers at her younger daughter’s school, reported that she appeared mentally disturbed, and frequently fantasized, once claiming that she had had an affair with Jack White, lead singer of The White Stripes.

The younger daughter’s phone was last active at 2:29 a.m., half an hour before she was found unconscious by her mother. During the examination of the scene of the elder sister’s death, numerous containers of cosmetics and cleaning products found in the bathroom were seized. Three years after the women’s deaths, an examination of the seized items revealed that one of them was a 500 g package of sodium nitrite (98.5% NaNO2). It turned out to have been ordered online two months before her death using the mother’s information. An analysis confirmed that the contents of the package were indeed sodium nitrite, with 434 g of the substance remaining. There were no DNA or fingerprint traces that could be assessed.

The younger sister’s life situation was then analyzed to determine whether she might have died by suicide or extended suicide. It was determined that while there were many factors indicating that she had died by suicide, there was no basis to assume extended suicide, i.e., that she had intended to kill her older sister and herself. The younger sister was described as gifted and artistically talented. She was calm, polite, sensitive, empathetic, delicate, and withdrawn. However, while she was generally well-liked by her peers, she did not form close relationships with them and had no close friends; she did not visit anyone, no one visited her and she did not attend parties. She was described as occupying her own world and was regarded as an outsider. She dressed in black and wore oversized clothes that hid her upper limbs: she wore long-sleeved blouses and lots of bracelets. Her friends commented that she was thin and pale, but no one knew why.

She was exceptionally mature and seemed to have taken over her mother’s household duties. Since the pandemic began, she had completely cut herself off from everything—she did not log in to remote classes, had more than 50% absences, and as a result, she had to take placement exams at the end of the school year, which she passed without issue. According to the case file, she attempted suicide a year earlier, allegedly taking her mother’s unspecified medication and tying a tie to the bathroom doorknob. After this attempt, she received no professional psychiatric (medication) or psychological (therapy) help. Since the pandemic began, she had only been to school twice, both days immediately before she died by suicide.

Ultimately, it was concluded that the younger sister died by suicide by using sodium nitrite, and the older sister died as a result of an accident: she had either eaten the vomit or the sodium nitrite from the backpack on the younger sister’s bed, either directly from the package or from the power spilt on the bedding.

## 3. Discussion

New methods to commit suicide have always been found, but the rapid development of the internet, social media, and instant messaging has accelerated the dissemination of these methods and placed them increasingly within the reach of a wider audience. Among them, young people are particularly vulnerable, being frequently online and at increasing risk of mental health problems, such as depression, anxiety disorders, eating disorders, and addictions [14].

Advice on how to successfully commit suicide may appear in a post on social media, from where it may be quickly shared to other formats in other languages and to other countries. Unfortunately, this can often end with the use of the new method, which may result in an unsuccessful attempt and hospitalization in the best-case scenario, or death in the worst case.

One currently popular method is acute sodium nitrite poisoning, which was initially included in a 2006 book describing various methods of suicide. Later, the method was disseminated to a wider audience through the internet, and occasionally cases have been encountered by clinical toxicologists, forensic pathologists and laboratory toxicologists. Its popularity can be attributed to its ready availability, and low price. The title of the book has been deliberately omitted from this paper to avoid the so-called Werther effect [15].

Both the diagnosis of sodium nitrite poisoning in living individuals and the determination of the cause of death can be challenging. While autopsy findings in some cases reveal a characteristic chocolate-colored livor mortis and blood [10], suggestive of nitrite poisoning, which can guide laboratory tests, this is not always the case: such coloration is typically noted at higher concentrations of MetHb, and death can also occur at lower levels [16]. Similarly, if residue is found at the scene in a container labeled as containing sodium nitrite, it is sufficient to send the postmortem material for testing; however, the container may not be found due to it being removed by either the suicide victim or someone else, such as a family member, due to exclusionary life insurance clauses. Therefore, if postmortem lividity and blood are normal in color and sodium nitrite is not found at the scene, poisoning may not be suspected and there is little chance that appropriate toxicology tests will be ordered. In such situations, it is therefore unlikely that the correct cause of death will be determined. Although, some countries with better-funded forensic programs may employ toxicology panels broad enough to cover most substances that can cause death.

Nitrite poisoning can be detected by various methods including gas chromatography, liquid chromatography, ion chromatography (IC) [17], the Griess method, and sometimes only strip tests that detect the presence of these compounds in urine [18]; in addition, new testing methods for nitrite poisoning are being identified, as well as new biological materials. For example, an Italian study describes the possibility of determining nitrites and nitrates in biological fluids using capillary ion analysis (CIA) with UV detection [19]. A Polish study based on the Griess method, an old method of detecting nitrite ions, found that both rib cartilage and vitreous humor can serve as alternative evidence in sodium nitrite poisoning [20]. In turn, a South Korean study [21] evaluated the potential of ion chromatography combined with a conductivity detector to quantify nitrite and nitrate levels in pericardial fluid and cerebrospinal fluid, both materials that are not usually used for research.

Some studies of nitrite poisoning focus on MetHb detection methods. A Turkish study on the postmortem detection of MetHb using MRI found that the presence of MetHb results in a bright (hyperintense) blood signal in the T1 sequence, both in intracardiac blood from chest X-ray and in blood samples in test tubes [3]. However, it should be remembered that MetHb levels may vary significantly in fatal cases and should not be used as the sole criterion for diagnosing death caused by sodium nitrite: MetHb levels may be substantially elevated, only slightly elevated, or even normal [22,23,24]. It should also be noted that even with high methemoglobinemia (73–92.7%), the patient may survive if the poisoning is recognized quickly and methylene blue is administered promptly [3,25]. In addition, spontaneous overtime methemoglobin formation may occur after death [26].

The case of double poisoning described herein presents another potential diagnostic challenge: determining the manner of death. Generally speaking, the manner of death is classified as natural, accident, suicide, homicide, and undetermined; among these, poisoning can be any of the last four. In the scientific literature on nitrite poisoning, the most frequent causes are given as accidental and suicidal poisonings. Accidental sodium nitrate poisonings were either culinary in nature (too much sodium nitrite used in the production of cold cuts, confusing sodium nitrite packaging with sugar or salt) or pharmacological in nature, e.g., improper preparation of the drug by the pharmacist [3,4,18,24,27]. Suicides have also been reported in both men and women, young and old [9,16,18,22,25,28,29,30,31,32,33,34,35,36]. No paper reporting the use of nitrite use in homicide could be found while researching the present study, although this does not necessarily mean that no such cases have occurred.

Once the cause and mechanism of death have been determined, a crucial part of the investigation is to determine the manner of death. In some cases, this is very easy: the circumstances of death may directly indicate a mistake, e.g., during the production of cold cuts, or suicide, e.g., a suicide note left behind, computer or cell phone data, or a history of previous suicide attempts and/or psychiatric treatment for depression or other mental disorders. Sometimes, however, the situation is more complicated, as demonstrated by the present case. The family’s life situation suggested that due to so-called life difficulties, addictions, financial problems, and childcare issues, especially with the older, profoundly autistic daughter, the mother may have decided to kill her daughters. The mother herself provided alcohol and drugs to the younger daughter; she also failed to adequately care for the physical well-being, failing to provide food at home, ostensibly due to poverty, yet had money for alcohol and drugs, and to support the mental well-being of her children, by failing to react after the younger daughter’s previous suicide attempt. Those around the mother believed that she was mentally disturbed and should seek psychiatric treatment. Suspicions that the mother was the perpetrator were strengthened by the fact that the sodium nitrite was ordered using her personal information. It was also possible that when the younger daughter died by suicide, the mother, out of desperation, killed the other. Furthermore, the extended suicide scenario also seemed plausible in the initial stages of the investigation; the friends of the younger sister described her concern for her older sister and her understandable fear of what would happen to her when left alone with her mother.

Ultimately, after analyzing the extensive evidence and expert opinions, the most likely course of events is believed to be suicide or accident. However, this might not have been possible if the investigators had not approached the case so meticulously and comprehensively: the proceedings lasted for four years. In such cases, the actual perpetrators may well remain undetected, or even an innocent party may be accused, which, as highlighted by the work of the Innocence Project, is more common than one might think.

## Figures and Tables

**Table 1 ijms-27-00218-t001:** Test results of the younger sister. The blood was not tested for nitrites because it had already been used up in earlier analysis.

Sample	NO_2_^−^ [µg/g]	Nitrite Content Expressed as NaNO_2_ [µg/g]	NO_3_^−^ [µg/g]
Stomach	6.8	10.1	not detected
Liver	1.3	2.4	not detected
Intestines	12.5	18.8	not detected
Kidney	2.1	3.2	not detected

**Table 2 ijms-27-00218-t002:** Test results of the older sister.

Sample	NO_2_^−^ [µg/g]	Nitrite Content Expressed as NaNO_2_ [µg/g]	NO_3_^−^ [µg/g]
Blood	0.8	1.3	not detected
Stomach	5.2	7.8	not detected
Liver	4.3	6.5	not detected
Intestines	8.1	12.1	not detected
Kidney	not detected	-	not detected

**Table 3 ijms-27-00218-t003:** Vomit test results.

Sample	NO_2_^−^ [µg/g]	Nitrite Content Expressed as NaNO_2_ [µg/g]	NO_3_^−^ [µg/g]
Vomit	94.5	141.7	<1.5

## Data Availability

The original contributions presented in this study are included in the article. Further inquiries can be directed to the corresponding author.

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
