# Peer review of "Double Fatal Sodium Nitrite Poisoning—Double Homicide, Extended Suicide, or Suicide and Accident?"

_ijms, 2025, doi:10.3390/ijms27010218_

Round 1

Reviewer 1 Report

Comments and Suggestions for Authors This case report discusses the double fatal sodium nitrite poisoning of two sisters, aged 17 and 31, and explores the possible manners of death: double homicide, extended suicide, or suicide and accident. 

The topic of the manuscript is not directly aligned with the focus of the Special Issue “Advances in Post-Mortem Toxicology.” Its emphasis is primarily on the forensic and judicial aspects rather than on toxicology per se or on the interpretation of toxicological results. In terms of novelty, it does not offer substantial advancements within the field of forensic toxicology.

Overall, because it addresses a highly relevant subject—specifically the documented rise in suicides by nitrite ingestion—it is a topic worthy of publication. However, the manuscript requires substantial improvement in the authors’ use of English, and the abstract in particular should be thoroughly revised.

Comments on the Quality of English Language

As mentioned above, the manuscript requires substantial improvement in the authors’ use of English, and the abstract in particular should be thoroughly revised.

Author Response

Comments 1: The manuscript requires substantial improvement in the authors’ use of English, and the abstract in particular should be thoroughly revised.

Response 1: Thank you for pointing this out. We agree with this comment. Therefore, we have made a language revision.

Reviewer 2 Report

Comments and Suggestions for Authors

The author describe an interesting cases arising concerns about sodium nitrite, I think it is a well done paper and can be published in the present form.

Author Response

Comments 1: The author describes an interesting cases arising concerns about sodium nitrite, I think it is a well done paper and can be published in the present form.

Response 1: Thank you very much for your positive review.

Reviewer 3 Report

Comments and Suggestions for Authors

Dear Author(s)

I have read your manuscript with interest and have very few comments to make.

Since testing for sodium nitrite is not routine in many laboratories, it is essential to consider that suspicion may arise from the discovery of the substance at the scene of death. From the description in the case report, it appears that sodium nitrite was found after tests identified the substance in the two victims, who did not show any typical signs of poisoning. Is that correct? Or was the package of sodium nitrite discovered in the backpack during the police investigation, which prompted a direct search for the substance?

Apart from this point, due to a probable layout error, Table 2, which relates to the toxicological results of the older sister, has been relocated to another position within the text.

In Table 1 relating to the younger sister, however, the blood analysis is not reported. If, in this case, the sodium nitrite analysis was not performed on blood, I would explain why in the text.

Author Response

Comment 1: From the description in the case report, it appears that sodium nitrite was found after tests identified the substance in the two victims, who did not show any typical signs of poisoning. Is that correct? Or was the package of sodium nitrite discovered in the backpack during the police investigation, which prompted a direct search for the substance?

Response 1: Thank you for your important remark. In this case, nitrates were detected due to a screening test. There was a package containing sodium nitrate in the victim’s backpack, but unfortunately, it was revealed 3 years after sister's death. An adequate explanation has been included in the test and highlighted in yellow.

Comment 2: Due to a probable layout error, Table 2, which relates to the toxicological results of the older sister, has been relocated to another position within the text. 

Response 2: Thank you for your mindful remark. Indeed, it was a layout error, and I have improved it.

Comment 3: In Table 1, relating to the younger sister, however, the blood analysis is not reported. If, in this case, the sodium nitrite analysis was not performed on blood, I would explain why in the text.

Response 3: Thank you for pointing this out. In the case of the younger victim, sodium nitrate analysis was not performed on the blood because the blood sample had already been used up in earlier analysis. This information was added to Table 1 description.